Association between intrinsic disorder and serine/threonine phosphorylation in Mycobacterium tuberculosis

Singh Gajinder Pal gajinder.pal.singh@gmail.com
School of Biotechnology, KIIT University , Patia, Bhubaneswar, Odisha , India
Uversky Vladimir
Electronic publication date: 2015 Jan 8
Publication date: 2015
Volume: 3
Electronic Location ID: e724
Received 2014 Oct 29; Accepted 2014 Dec 21
Copyright: © 2015 Singh
Copyright year: 2015
Copyright holder: Singh
License: This is an open access article distributed under the terms of the Creative Commons Attribution License, which permits unrestricted use, distribution, reproduction and adaptation in any medium and for any purpose provided that it is properly attributed. For attribution, the original author(s), title, publication source (PeerJ) and either DOI or URL of the article must be cited.
License URL: https://creativecommons.org/licenses/by/4.0/

Keywords: Protein disorder, Intrinsic disorder, Phosphorylation, Serine/threonine phosphorylation, Mycobacterium tuberculosis

Funding: Science and Engineering Research Board SB/YS/LS-256/2013 Funding was provided by the Science and Engineering Research Board (SERB), Department of Science and Technology, Govt. of India - NO. SB/YS/LS-256/2013. The funders had no role in study design, data collection and analysis, decision to publish, or preparation of the manuscript.

==============================
Serine/threonine phosphorylation is an important mechanism that is involved in the regulation of protein function. In eukaryotes, phosphorylation occurs predominantly in intrinsically disordered regions of proteins. Though serine/threonine phosphorylation and protein disorder are much less prevalent in prokaryotes, some bacteria have high levels of serine/threonine phosphorylation and disorder, including the medically important M. tuberculosis. Here I show that serine/threonine phosphorylation sites in M. tuberculosis are highly enriched in intrinsically disordered regions, indicating similarity in the substrate recognition mechanisms of eukaryotic and M. tuberculosis kinases. Serine/threonine phosphorylation has been linked to the pathogenicity and survival of M. tuberculosis. Thus, a better understanding of how its kinases recognize their substrates could have important implications in understanding and controlling the biology of this deadly pathogen. These results also indicate that the association between serine/threonine phosphorylation and disorder is not a feature restricted to eukaryotes.

Introduction

The reversible phosphorylation of serine and threonine residues is a widespread post-translational modification in eukaryotes, with more than a third of proteins phosphorylated during their lifetime (Albuquerque et al., 2008). Phosphorylation can modify protein interactions, enzyme functions, localization and degradation. Although regulation and signal transduction in bacteria were traditionally thought to be mediated by histidine and aspartate phosphorylation in two-component systems, the occurrence and importance of phosphorylation of serine/threonine (S/T) residues has recently gained much attention (Cousin et al., 2013; Kobir et al., 2011; Mijakovic & Macek, 2012). Large-scale mass spectrometry based analyses have revealed S/T phosphorylation in a number of bacteria (Mijakovic & Macek, 2012).

One of the most interesting findings concerning S/T phosphorylation in eukaryotes is its association with intrinsically disordered regions (Amoutzias et al., 2012; Collins et al., 2008; Gnad et al., 2009; Gsponer et al., 2008; Iakoucheva et al., 2004; Marchini et al., 2011). Intrinsically disordered regions lack a well-defined three-dimensional structure and are characterized by low hydrophobic amino acid content and a high net charge (Uversky, Gillespie & Fink, 2000). These characteristic physiochemical properties allow for accurate predictions of disordered regions across proteomes (Monastyrskyy et al., 2014). Disordered regions are often associated with the ability to bind to multiple partners in a transient manner (Dunker et al., 2005; Fink, 2005; Patil & Nakamura, 2006; Singh, Ganapathi & Dash, 2007; Tompa, Szasz & Buday, 2005; Wright & Dyson, 1999). These regions may undergo a disorder to order transition upon binding, with a decrease in conformational entropy. This process uncouples binding affinity and specificity, thus allowing highly specific interactions to be reversible (Dyson & Wright, 2005; Tompa, 2002). It has been proposed that disordered regions are ideally suited for regulation by reversible phosphorylation due to their high surface accessibility and transient mode of interactions (Collins et al., 2008; Dyson & Wright, 2005; Iakoucheva et al., 2004; Tompa, 2002). Disordered regions are highly abundant in eukaryotes, with approximately one third of proteins predicted to have at least one long (>30 residues) disordered region and approximately 19% of residues predicted to be in a disordered state (Ward et al., 2004). In contrast, most bacteria have much less disorder in their proteome, with approximately 4% proteins predicted to contain long disordered regions and approximately 6% of residues predicted to be disordered (Ward et al., 2004). The association between disorder and S/T phosphorylation has not been investigated in prokaryotes.

Among bacteria, S/T phosphorylation is the most well studied in M. tuberculosis and is linked to its survival, pathogenesis and virulence (Av-Gay & Everett, 2000; Cousin et al., 2013; Pereira, Goss & Dworkin, 2011). This bacterium shows one of the highest rates of phosphorylation among studied bacteria, with 8% of its proteins identified as phosphorylated (Prisic et al., 2010). M. tuberculosis also has a high disorder content, with approximately 10% disordered residues (Ward et al., 2004). The prevalence and importance of S/T phosphorylation in M. tuberculosis prompted the question of whether the association observed between S/T phosphorylation and disorder in eukaryotes might be present in M. tuberculosis, which indeed was found to be the case. Furthermore, this association was also identified in other bacteria.

Materials and Methods

Data on M. tuberculosis phosphoproteins and phosphosites were obtained from Prisic et al. (2010). This study identified 301 phosphoproteins and 500 S/T phosphorylation events. For 215 of these sites, the specific residue that was modified could be identified with high confidence. The M. tuberculosis proteome was obtained from the Tuberculist database (Lew et al., 2013). For disorder prediction, I utilized the IUPred method (Dosztanyi et al., 2005a). This method is based on the observation that disordered regions do not form sufficiently favorable interactions to fold and thus have high estimated energy content (Dosztanyi et al., 2005b). I also utilized the ESpritz program (Walsh et al., 2012), which is conceptually different from IUPred. This method is a machine-learning based predictor that was trained on experimentally characterized disordered regions (missing regions in X-ray structures in PDB). I also used the MFDp2 disorder prediction tool, which is an ensemble disorder prediction tool (Mizianty, Peng & Kurgan, 2013; Mizianty, Uversky & Kurgan, 2014). Secondary structure prediction was performed at the Network Protein Sequence Analysis (NPSA) server (Combet et al., 2000) using a consensus approach (Deleage, Blanchet & Geourjon, 1997). To analyze conservation of S/T sites, 14 diverse mycobacterium species were chosen (M. intracellulare, M. smegmatis, M. chubuense, M. avium, M. gilvum, M. abscessus, M. marinum, M. bovis, M. canettii, M. kansasi, Mycobacterium sp. MCS, Mycobacterium sp. JLS, Mycobacterium sp. KMS, and Mycobacterium sp. JDM60). Orthologs of M. tuberculosis in mycobacteria were identified using the reciprocal best blast approach (Wolf & Koonin, 2012), and aligned using Clustal Omega (Sievers & Higgins, 2014). Alignment positions with gaps were excluded from the analyses. Of the 215 sites, 139 sites were present in proteins which had orthologs in all other 14 species. Of these 139 sites, 103 sites were without gaps. Positions with replacement of serine with threonine and vice-versa were considered to be conserved. The number of species in which S/T residues were present at the alignment position was calculated as a measurement of conservation. Phosphosite data for other bacteria were obtained from respective publications (Aivaliotis et al., 2009; Lin et al., 2009; Macek et al., 2007; Manteca et al., 2011; Misra et al., 2011; Parker et al., 2010; Soufi et al., 2008; Yang et al., 2013).

Results

Mass spectrometry based analysis previously revealed 301 phosphoproteins in M. tuberculosis containing 500 S/T phosphorylation sites (Prisic et al., 2010). First, I tested whether phosphoproteins in M. tuberculosis were more likely to be disordered (i.e., have long (≥30 residues) disordered regions). I utilized the IUPred program to predict disordered regions at the proteome wide level (Dosztanyi et al., 2005b). Phosphoproteins were approximately twice as likely to be disordered compared to non-phosphoproteins (29.6% vs. 13.4%, respectively; Fisher test p 4e-12). Because longer proteins are also more likely to have long disordered regions, I tested whether phosphoproteins have higher percentage of disordered residues. Indeed, phosphoproteins have higher percentage of disordered residues than do non-phosphoproteins (16.7% vs. 12.0%, respectively; two tailed t-test p 3e-5).

Of the 500 phosphorylation events detected in M. tuberculosis, the phosphoresidues could be identified for 215 sites with high confidence (Prisic et al., 2010). For these sites, I tested whether phosphorylated S/T (pS/T) residues were more likely to be disordered compared to non-phosphorylated S/T (npS/T) residues from the same set of proteins. Overall, 39.1% of the pS/T sites were disordered compared with 22.4% of npS/T sites (Fisher test p 6e-8; Fig. 1). The results were very similar when another disorder prediction method, ESpritz (Walsh et al., 2012), was used (52.6% pS/T disordered compared with 27.8% npS/T sites; Fisher test p 8e-14, Fig. 1). The more recently described disorder predictor MFDp2 (Mizianty, Peng & Kurgan, 2013; Mizianty, Uversky & Kurgan, 2014) also gave similar results (43.7% pS/T disordered compared with 19.9% npS/T sites; Fisher test p 6e-15; Fig. 1). Disordered regions are also characterized by high irregular secondary structure regions (i.e., coil regions). Thus, I tested whether pS/T residues were enriched in coil regions of the proteins. pS/T residues were more likely to occur in predicted coil regions than were npS/T residues (70.2% pS/T sites in coils compared with 55.7% npS/T sites in coils; Fisher test p 2e-5; Fig. 2). A depletion of pS/T residues in beta-sheet regions was also observed (4.2% pS/T sites in sheet compared with 11.3% npS/T sites in sheets; Fisher test p 4e-4; Fig. 2), whereas no significant difference was found for helix regions (22.8% pS/T sites in helices compared with 27.1% npS/T sites in helices; Fisher test p 0.2; Fig. 2).

Figure 1 Phosphorylated serine/threonine sites in M. tuberculosis are more likely to be disordered.

Both phosphorylated and non-phosphorylated serine/threonine sites are from the same set of proteins. The disorder was predicted using the IUPred, ESpritz and MFDp2 methods. The Fisher test p values were 6e-8, 8e-14 and 6e-15, respectively. pS/T-phosphorylated serine/threonine, npS/T- non-phosphorylated serine/threonine. Error bars indicate 95% confidence intervals of the mean from 1,000 bootstrap samples.

Figure 2 Association between predicted secondary structure and phosphorylation of serine/threonine sites in M. tuberculosis.

Phosphorylated serine/threonine sites occur preferentially in coil regions, are significantly depleted in sheet regions and show no significant difference in helix regions. The Fisher test p values are 2e-5, 4e-4 and 0.2, respectively. pS/T-phosphorylated serine/threonine, npS/T- non-phosphorylated serine/threonine. Error bars indicate 95% confidence intervals of the mean from 1,000 bootstrap samples.

Next, I tested whether M. tuberculosis pS/T and npS/T differed in the conservation across mycobacteria. I identified orthologs of M. tuberculosis among 14 mycobacterial species, aligned their sequences and calculated the ratio of conservation of pS/T sites with npS/T sites for each phosphoprotein (see methods). In 65% (49/76) of the proteins, pS/T sites were more conserved than npS/T sites from the same protein (Fig. S1). This proportion was significantly different from the expected value of 50% (Binomial test p 0.008). Because disordered regions and disordered pS/T sites are known to evolve faster (Brown et al., 2002; Landry, Levy & Michnick, 2009), I analyzed disordered and ordered sites separately. In 69% (40/58) of the proteins, ordered pS/T sites were more conserved than ordered npS/T sites (Binomial test p 0.003), whereas in 61% (14/23) of the proteins, disordered pS/T sites were more conserved than disordered npS/T sites (Binomial test p 0.2). It is likely that the lack of higher conservation of disordered pS/T sites might be due to the low number of sites and proteins analyzed.

Prisic et al. (2010) conducted in vitro phosphorylation of 13-mer synthetic peptides corresponding to in vivo phosphorylation sites using different purified kinases. They could find phosphorylation of approximately half of these peptides. Based on these results, I tested whether different kinases have differential preferences for predicted disordered phosphoacceptors and found that PknA has a slightly higher preference for disordered phosphoacceptors compared with other kinases (Fig. S2). However, the uncorrected Chi-square test p-value was only 0.04, whereas other kinases showed uncorrected p-values >0.05.

Mass spectrometry has been used to identify S/T phosphorylation sites in a number of prokaryotes other than M. tuberculosis (Mijakovic & Macek, 2012). Finally, I tested the association between disorder and phosphorylation in other prokaryotes, and found that an association between disorder and phosphorylation was present in some prokaryotes but not in others (Table 1).

Table 1 Number of disordered and ordered serine/threonine sites in phosphoproteomes of different bacteria.a

Organism	Number of
localized pST	% pST
disordered	% npST
disordered	Fisher test
p-value	% proteome disordered	
Mycobacterium tuberculosis	215	39.1	22.4	6E-08	11.7	
Escherichia coli	97	8.2	5.3	0.25	5.2	
Bacillus subtilis	92	4.3	8.7	0.18	5.7	
Thermus thermophilus	42	11.9	4.2	0.04	4.9	
Streptomyces coelicolor	20	60.0	36.8	0.06	18.6	
Streptomyces coelicolor *	211	64.5	40.0	3E-12	18.6	
Klebsiella pneumoniae	44	11.4	9.9	0.80	4.9	
Lactobacillus lactis	66	18.2	15.3	0.49	5.0	
Synechococcus sp.	354	10.7	12.9	0.26	6.8	
Halobacterium salinarum	73	64.4	38.7	2E-05	20.9	
Listeria monocytogenes	120	11.7	13.5	0.68	5.0	
Mycoplasma pneumoniae	65	9.2	16.4	0.16	9.0	
Streptococcus pneumoniae	147	12.2	14.0	0.63	5.0	
Notes.

* S. coelicolor appears twice because of two independent studies on S/T phosphorylation.

a Disorder predicted using the IUPred method.

Discussion

Here, I show the enrichment of localized S/T phosphosites in disordered regions of proteins in M. tuberculosis. pS/T sites in M. tuberculosis are approximately 2 fold more likely to occur in disordered regions compared with ordered regions (Fig. 1). This preference is similar to that observed in eukaryotes, where pS/T sites are 2-to-3 fold more likely to occur in disordered regions (Amoutzias et al., 2012; Landry, Levy & Michnick, 2009; Marchini et al., 2011). However, because the percentage of disordered residues is much higher in eukaryotes than in M. tuberculosis, approximately 80–90% of pS/T sites in eukaryotes occur in disordered regions (Amoutzias et al., 2012; Collins et al., 2008; Landry, Levy & Michnick, 2009; Marchini et al., 2011) compared to approximately 40% in M. tuberculosis (Fig. 1). The association between protein disorder and phosphorylation may offer similar advantages as those proposed in eukaryotes, including binding to multiple partners and transient mode of interaction (Dunker et al., 2005; Dyson & Wright, 2005; Fink, 2005; Iakoucheva et al., 2004; Patil & Nakamura, 2006; Singh, Ganapathi & Dash, 2007; Tompa, 2002; Tompa, Szasz & Buday, 2005; Wright & Dyson, 1999), which are a prerequisite for regulatory interactions. Thus, M. tuberculosis and eukaryotic S/T phosphorylation dependent regulation may be more similar than generally appreciated. Whereas most bacteria have a low amount of protein disorder, M. tuberculosis has a high disorder content. The high disorder content in M. tuberculosis may possibly allow higher levels of S/T phosphorylation. Some other bacteria with high disorder content (Streptomyces coelicolor and Halobacterium salinarum) also show an association between disorder and S/T phosphorylation, whereas bacteria with low disorder contents do not (Table 1). The exception is Thermus thermophilus, which exhibited an enrichment of phosphorylation in disordered regions with marginal statistical significance (Table 1). Previously, a large fraction of phosphosites in Thermus thermophilus was observed in loop regions (Takahata et al., 2012), though no statistical test for enrichment was performed. It would be interesting to study S/T phosphorylation in other high disorder-containing bacteria to test whether these bacteria also show high levels of S/T phosphorylation.

PknA has a slightly higher preference for disordered phosphoacceptor sites on synthetic 13-mer substrate peptides under in vitro conditions than other kinases (Fig. S2). However, these results should be taken with the caveat that the structure of a peptide under in vitro conditions might be very different from the in vivo structure in the context of the full protein. Thus, the differential preferences of kinases towards disordered substrates under in vivo conditions remains an open question.

In eukaryotes, the conservation of pS/T sites has been a matter of some debate, with some studies reporting no higher conservation of pS/T sites (Gnad et al., 2009; Landry, Levy & Michnick, 2009), while others reported higher conservation of pS/T sites (Chen, Chen & Li, 2010; Gray & Kumar, 2011). More recently, it was argued that pooling conservation rates from multiple proteins may bias the results, since this approach does not account for the large differences in the conservation of different proteins. Thus, the conservation of pS/T sites should be compared with the conservation of npS/T sites from the same protein (Gray & Kumar, 2011). Indeed, in M. tuberculosis, there was no significant difference in the conservation of pS/T and npS/T sites among mycobacteria, when their averages were compared across proteins (mean conservation in 11.34 and 11.28 species out of 14, respectively; Wilcox test p 0.4). However, when the conservation of pS/T was compared with that of npS/T from the same protein, pS/T sites were found to be more conserved relative to npS/T sites. It might be useful to prioritize pS/T sites with high conservation (relative to npS/T sites from the same protein) for further experimental studies. Phosphoproteomic analyses on more mycobacteria would also be highly valuable to identify S/T sites phosphorylated in multiple mycobacteria. In eukaryotes, disordered pS/T sites demonstrate higher rate of evolution (Landry, Levy & Michnick, 2009). In M. tuberculosis, I did not find a statistically significant difference in the conservation of disordered pS/T sites, compared with disordered npS/T sites. However, due to the low number of sites and proteins analyzed, this issue would need to be revisited when more data become available.

Further important questions for the future include the following: (1) Are disordered and ordered S/T phosphosites functionally different? and (2) Do different kinases differ in their preferences for disorder in their substrates under physiological conditions? Incorporating disorder information might also be useful for the prediction of novel S/T phosphosites (Miller et al., 2009), as has been shown in eukaryotes (Iakoucheva et al., 2004; Neduva et al., 2005). S/T kinases and their substrates have been linked to the survival, pathogenesis and virulence of M. tuberculosis (Av-Gay & Everett, 2000; Cousin et al., 2013; Pereira, Goss & Dworkin, 2011). Thus, these finding may facilitate an understanding of the basic biology of this deadly pathogen.

Supplemental Information

Figure S1 Frequency distribution of relative evolutionary rate of pS/T

Click here for additional data file.

Figure S2 Percentage of disordered phosphoacceptors are shown for different kinases

The phosphorylation was carried out by purified kinases under in vitro conditions on synthetic 13-mer peptides corresponding to in vivo phosphorylation sites (Prisic et al., 2010). Uncorrected Chi-square p values were significant (<0.05) only for PknA.

Click here for additional data file.

I would like to thank Dr. Shampa Ghosh from Bionivid Technology, and Dr. Rahul Modak and Dr. Avinash Sonawane from KIIT University for critical reading of the manuscript.

Additional Information and Declarations

Competing Interests

Author Contributions

The author declares there are no competing interests.

Gajinder Pal Singh conceived and designed the experiments, performed the experiments, analyzed the data, contributed reagents/materials/analysis tools, wrote the paper, prepared figures and/or tables, reviewed drafts of the paper.

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
