# Peer review of "Association between intrinsic disorder and serine/threonine phosphorylation in Mycobacterium tuberculosis"

_PeerJ, doi:10.7717/peerj.724_

## Round 0.1 · original submission · Major Revisions

· Academic Editor

Major Revisions

Please pay close attention to all the critical points raised by both reviewers. Please also note that one of the prerequisites for publication in PeerJ is the requirement that the article is presented in an intelligible fashion and is written in standard English. PeerJ does not copyedit accepted manuscripts, so the language in submitted articles must be clear, correct, and unambiguous. Since reviewers indicated that the language of your paper is difficult to understand and includes many errors, I strongly encourage you to seek independent editorial help before submitting a revision. These services can be found on the web using search terms like "scientific editing service" or "manuscript editing service."

Reviewer 1 ·

Basic reporting

Except for the typographical/grammatical errors listed below, the manuscript is clearly written and adheres to PeerJ’s standards:
• Line 53 – Species name must be italicized.
• Line 62 – The sentence should read ‘The IUPred method …’
• Line 65 – The sentence should read ‘This method is a machine learning …’
• Line 67 – The sentence should read ‘… used the MFDp2 …’
• Line 70 – The sentence should read ‘… using a consensus approach …’
• Line 71 – The sentence should read ‘ … loop regions, I used a randomization …’
• Line 73 – The sentence should read ‘… out of a total of …’
• Line 77 – ‘genus’ should be pluralized to ‘genuses’
• Line 128 – Delete ‘the’ before ‘whether’
• Line 160 – The sentence should read ‘…for the future …’
• Line 167 – The sentence should read ‘… of this deadly pathogen.’
• Fig. 1 – Incorrect spelling for ‘Espritz’ in caption and figure
• In Table S1, can you clarify why there are two rows for S. coelicolor? Is this a typo?
• In lines 33-34, several citations have been included to support the statement. However, Landry et al. has been incorrectly cited for this statement. In fact, an additional citation should be included:

Gsponer, Jörg, et al. "Tight regulation of unstructured proteins: from transcript synthesis to protein degradation." Science 322.5906 (2008): 1365-1368.

The author is advised to include only those citations that first suggest and/or confirm the association between phosphorylation and intrinsic disorder.

Experimental design

Overall, the experimental design looks good. In particular, the author has taken care to ensure that biases from disorder predictors do not affect the final results. However, there are three concerns:
• The choice of M. tuberculosis for all the experiments needs to be motivated more clearly. The author states that the presence of high disorder and high pS/T content determined the choice of species. However, as the author points out in the supplementary table, there are other species that have both high disorder content and levels of phosphorylation. On a related note, there are other bacterial data sets as well: van Noort, Vera, et al. "Cross‐talk between phosphorylation and lysine acetylation in a genome‐reduced bacterium." Molecular systems biology 8.1 (2012).
• It is not clear why a randomization test was chosen to assign statistical significance to the proportion of phosphosites in disordered regions when a simple Chi-square or Fisher’s exact test would have sufficed. In fact, the author has done this in supplementary table 1 and an additional test seems redundant. It is suggested that the author either clarify the need for this test or remove the redundancy by reporting the P-value from the Fisher test.
• There are a few important issues regarding the conservation analysis that concern me:
1. Different proteins evolve at different rates and a simple fraction of conserved residues at a given position does not take this into account. The conservation at a position in the protein needs to be normalized with respect to the entire protein’s conservation. Gray and Kumar address this in their 2011 paper.
2. While the conservation levels of pS/T and S/T are not significantly different, they seem to be highly conserved in general (in ~5 out of the 7 species considered). This is probably due to the choice of species. I am not aware of divergence times in this genus, but the farther back in time you go, a clearer signal is likely to emerge. In fact, the studies cited by the author considered long evolutionary distances in eukaryotes. An alternative would be to use an unbiased approach by performing a reciprocal BLAST search against the entire nr database.
3. D and E residues are considered to be phosphomimetic residues [Pearlman, Samuel M., Zach Serber, and James E. Ferrell Jr. "A mechanism for the evolution of phosphorylation sites." Cell 147.4 (2011): 934-946.] and positions with changes to or from these residues should ideally be excluded.

Validity of the findings

• The findings from the conservation analysis need to be placed in the context of previous work. In eukaryotes, the question of whether pS/T are more conserved than S/T has been debated for a while now. While the papers cited in line 119 do show that they are indeed more conserved, but work from Landry et al. shows no difference in evolutionary rates. This ambiguity underscores the importance of the above point on the conservation analysis.
• The statements in lines 135 and 141 on the similarity between M. tuberculosis and eukaryotic species in terms of phosphosite-disorder associations needs to be explained a bit more. Perhaps, the author can list out some proportion values for eukaryotic species to show that this is indeed the case. It would also be good if the author can comment on the results here in the context of similar previous works.
• While the statement in Line 143 is intended to be speculation, it is worded strangely. To me, it is saying, ‘Phosphosites in M. tuberculosis cannot be associated with intrinsically disordered regions without high amounts of intrinsically disordered regions in the proteome’. This is akin to saying that this species has a high number of pS/T, which was already established in the 'Introduction' section.

Additional comments

This paper addresses an interesting question through a well-reasoned computational approach. However, the conservation analysis needs to be done more carefully with the eukaryotic debate in mind. Finally, the author needs to improve the manuscript with respect to the minor and major issues with the writing, especially in the ‘Discussion’ section.

Reviewer 2 ·

Basic reporting

No comments.

Experimental design

No comments.

Validity of the findings

No comments.

Additional comments

The manuscript by Gajinder Pal Singh focuses on the observations that the proteome of M. tuberculosis has both (a) relatively high disorder content and (b) relatively high number of phosphoS/Ts, and establishes a relationship between the two, as has been previously shown for eukaryotes.

Overall, the manuscript is well written, but leaves me wondering if the author is missing an opportunity to make a stronger claim about disorder and pS/Ts in bacteria? Suppl. Table S1 and discussion (lines 144-6) list two bacteria in addition to the M. tuberculosis where % disorder tracks with significant enrichment of disordered pS/T sites. Are the species in Suppl. Table S1 all for which there are experimental phos sites? Do predicted pS/T sites track well with experimental sites in these species? If so, doing a large scale predictive experiment might make sense.

Also left unexplored is the relationship between pS/T and pH/D in these bacterial species. Are pS/T sites regulatory elements in addition or a substitution for pH/D? Is there a general trend that with % disorder in a bacterial proteome, % pS/T goes up, and % pH/D goes down?

Other:

In Table S1, Streptomyces coelicolor appears twice?

In Figures 1 and 2, in order to provide a sense of variance for % disordered and % S/T sites, the author should include error bars. One way to estimate variance would be bootstrap sampling.

---

## Round 0.2 · accepted · Accept

· Academic Editor

Accept

Thank you very much for the careful consideration of the reviewers' comments and for the great job you did revising the manuscript.